# Real-Time Musculoskeletal Kinematics and Dynamics Analysis Using Marker- and IMU-Based Solutions in Rehabilitation

**DOI:** 10.3390/s21051804

**Published:** 2021-03-05

**Authors:** Dimitar Stanev, Konstantinos Filip, Dimitrios Bitzas, Sokratis Zouras, Georgios Giarmatzis, Dimitrios Tsaopoulos, Konstantinos Moustakas

**Affiliations:** 1Institute of Bioengineering, École Polytechnique Fédérale de Lausanne, 1018 Lausanne, Switzerland; 2Department of Electrical and Computer Engineering, University of Patras, 26504 Patras, Greece; filip.k@ece.upatras.gr (K.F.); bintzas@ece.upatras.gr (D.B.); zouras@ece.upatras.gr (S.Z.); ggiarmatzis@ece.upatras.gr (G.G.); moustakas@ece.upatras.gr (K.M.); 3Institute for Bio-Economy and Agri-Technology, Center for Research and Technology Hellas, 38333 Volos, Greece; d.tsaopoulos@certh.gr

**Keywords:** real-time, musculoskeletal, kinematics, dynamics, muscle forces, joint reactions, ground reactions, inertial measurement units

## Abstract

This study aims to explore the possibility of estimating a multitude of kinematic and dynamic quantities using subject-specific musculoskeletal models in real-time. The framework was designed to operate with marker-based and inertial measurement units enabling extensions far beyond dedicated motion capture laboratories. We present the technical details for calculating the kinematics, generalized forces, muscle forces, joint reaction loads, and predicting ground reaction wrenches during walking. Emphasis was given to reduce computational latency while maintaining accuracy as compared to the offline counterpart. Notably, we highlight the influence of adequate filtering and differentiation under noisy conditions and its importance for consequent dynamic calculations. Real-time estimates of the joint moments, muscle forces, and reaction loads closely resemble OpenSim’s offline analyses. Model-based estimation of ground reaction wrenches demonstrates that even a small error can negatively affect other estimated quantities. An application of the developed system is demonstrated in the context of rehabilitation and gait retraining. We expect that such a system will find numerous applications in laboratory settings and outdoor conditions with the advent of predicting or sensing environment interactions. Therefore, we hope that this open-source framework will be a significant milestone for solving this grand challenge.

## 1. Introduction

Movement is essential for human well-being, and diseases that affect it can significantly deteriorate life quality. Advances in mobile sensors (Inertial Measurement Units (IMU)), video technologies (markerless Motion Capture (MOCAP)), and traditional marker-based systems are used for real-world assessment of movement [1,2]. The application of these methods in the analysis of healthy and pathological conditions can be crucial for clinical decision making [3,4]. Complimenting direct measurements using validated computational models can also provide new sources of information for training and rehabilitation purposes [5,6]. It is a common practice that such assessments are carried out offline (post-processing after a session) and in dedicated MOCAP laboratories. To this end, we aim to examine the possibility of estimating a multitude of kinematic and dynamic quantities using computational musculoskeletal models in real-time for both in- and out-of-the-lab conditions.

Different research groups have approached the problem of estimating the kinematics and dynamics of movement using musculoskeletal models. Notably, in [7] the authors laid the foundation and presented a system capable of assessing human movement and muscle function in real-time. Their predictions were compared against the gait2392 model [8] within the open-source modeling framework OpenSim [9,10]. Statistically significant differences were found in the output variables, which may not influence the conclusions in clinical settings as concluded by the original authors [11]. However, a proprietary solution may limit this system’s use and extent towards a quick adoption of new technologies (e.g., markerless tracking). To this end, we will present an open-source framework for carrying real-time calculations based on the OpenSim tool-chain that is well-adopted by the biomechanics community.

An open-source solution (rtosim), that relies on OpenSim, capable of performing Inverse Kinematics (IK) and Inverse Dynamics (ID) calculations was presented in [12]. Multi-thread implementation was proposed to increase the throughput of the system. The authors presented a real-time filter for rejecting noise and calculating the kinematics’ first and second derivatives. The influence of the filter’s cutoff frequency was further studied, and a methodology for determining its value was presented. In this study, we examine the role of real-time filtering more closely and how it can negatively affect dynamic calculations at later stages. We propose a filter capable of accurately estimating the kinematics’ first and second derivatives under noisy conditions. This results in real-time calculations of kinematics and dynamics that closely resemble the alternative offline analyses.

The same group also extended their work [13,14] in order to calculate muscle action and joint reaction loads in real-time using Electromyography (EMG)-driven methods [15,16]. These methods take into account surface EMG measurements to calibrate the musculoskeletal model by adjusting muscle-related parameters while satisfying joint moments about one or more Degrees of Freedom (DoF). This approach’s limitation is that surface EMG measurements can induce noise due to soft-tissue movement artifacts that can cause problems during calibration and use of the method. Besides, a superficial measurement is not an ideal representative of whole muscle function. This approach might also be limited by the number and choice of calibrated DoF. In our implementation, we decided to resolve the muscle redundancy through an optimization similar to [7] so that the system can find applications where EMG measurements might not be possible. However, it should be pointed out that optimization-based methods might fail to predict muscle activity in pathological conditions such as cerebral palsy or Parkinson’s disease [17].

In this work, we envision that the real-time system can be used not only in laboratory settings but also in out-of-the-lab conditions, where the kinematics might be measured through IMU or other markerless devices. With some minor modifications to the IK method, one can easily track both body orientations and marker positions. An important pre-requirement is that one can either measure or predict the user’s interactions with the environment (e.g., ground reactions or contacts with objects) to carry out further dynamic analyses. In particular, we focused on estimating the Ground Reaction Forces and Moments (GRF&M) during walking using only kinematics. There have been numerous studies [18,19,20,21,22,23,24] that focused on the solution to this challenge. Model-based estimation of these wrenches (a 9D vector containing the point, force, and moment) proved to be an ambitious goal under the context of any movement. This is because any small error in these estimates can lead to large variations in the joint moments, among other quantities. Our findings suggest that joint moments are sensitive to estimates of the GRF&M, but some moments can be determined accurately if one or two of the unknowns (ground reaction forces, Center of Pressure (CoP), or moments) is measured.

In the following sections, we will closely examine the challenges mentioned above. In Section 2.1, will extend the IK method to facilitate marker- and IMU-based tracking. We will continue with the real-time filtering and differentiation method (Section 3.1) that works very well under noisy measurements. In Section 2.3, we will examine the problem of predicting the GRF&M using only kinematic information. We will then present the formalization for real-time muscle redundancy optimization (Section 3.4). We will provide context related to the methods’ performance and outline techniques that reduce latencies in the calculations (e.g., parallelization of modules and estimation of muscle forces). In the results section, we will thoroughly examine the accuracy of the real-time calculations, comparing our findings with the offline methods to ensure that the former maintains the right balance between real-time applications’ constraints and accuracy. A use case scenario of the proposed framework is presented (Section 2.5) in the context of rehabilitation and gait retraining to reduce knee loads [6,25] (Section 3.5). Determining kinematics, joint moments, muscle forces, and joint reaction loads in real-time can find many applications in sports biomechanics, ergonomics, and rehabilitation. We expect that real-time calculations will be a viable solution in the near future, not only in laboratory settings but also in outdoor conditions with the advent of predicting or sensing environment interactions.

## 2. Methods

Figure 1 presents an overview of the real-time musculoskeletal kinematics and dynamics analysis pipeline. The time-wise execution is divided into the acquisition and processing threads that work independently while sharing results through a thread-safe data buffer. The acquisition part is connected with the MOCAP system that can work with marker- and IMU-based systems. Other types of sensor sources such as force plates, EMG, and pressure insoles can also be considered inputs. In the case of marker-based systems, we implement a marker completion step to solve the issue of missing markers due to occlusion [26]. The IK module is solved on the acquisition thread for each frame of data received, and its output (generalized coordinates) and other raw inputs are placed in the data buffer. Depending on the application, the processing thread can arrange the operations required to extract meaningful information from the musculoskeletal model in real-time. In this example, we present a workflow for calculating the kinematics, generalized forces, muscle forces, and joint reaction loads. An important aspect of this workflow is the real-time filtering and differentiation of the generalized coordinates obtained from IK, which is the input to ID and consequent blocks. The ID and joint reaction calculations are low-latency operations. However, determining the muscle forces that are necessary for estimating the joint reactions is a challenging task. The model calibration module refers to adjusting the musculoskeletal model to account for the subject’s anthropometric parameters using static trials and functional tasks. The internal implementation of the different building blocks will be explained in the following subsections.

### 2.1. Marker- and IMU-Based Inverse
Kinematics

The current implementation of IK can handle marker position and IMU orientation tracking. The implementation of IK solves an optimization problem that minimizes the holonomic (position) constraints, the weight marker, and orientation error
(1)$$\begin{array}{ccc} \underset{q}{\mathrm{minimize}}\; & \; & \frac{1}{2}q_{err}^{T}\left(q\right)q_{err}\left(q\right)+\frac{1}{2\sum w_{i}}\sum_{i}^{NM}w_{i}r_{i}^{2}\left(q\right)+\frac{1}{2\sum w_{j}}\sum_{j}^{NO}w_{j}\alpha_{j}^{2}\left(q\right) \end{array}$$
where, $q_{err}$ denotes the holonomic constraint error, $r_{i}$ the difference between experimental and virtual marker position, $\alpha_{j}$ the angular difference between experimental and virtual IMU orientation, and *w* are the weights (default is 1) associated with the different error terms. The superscript *T* over a vector or matrix denotes the transpose operation. Please note that this implementation can handle models that contain holonomic constraints due to the first term in the optimization that penalizes these errors.

Concerning IMU integration, a calibration procedure is defined to match the sensor orientations with the corresponding body segment orientations. The subject is asked to stand in a pose that matches the default pose of the virtual model for a few seconds. The sensor orientations ${}^{G_{IMU}}R^{S}$ are expressed in the Earth’s coordinate system. In order to express them in the global coordinate system of OpenSim, a transformation ${}^{G_{o}}R^{G_{IMU}}$ is first required. Next, a transformation $R_{heading}$ is applied which is computed from the angular difference between the orientation of the base sensor’s anterior axis (typically the IMU placed on the pelvis or torso) measured during the static trial, and the orientation of the anterior axis of the virtual model. This is necessary to compensate for the heading direction of the subject. The estimated transformation $R_{heading}{}^{G_{o}}R^{G_{IMU}}$ is considered constant throughout the session and can be applied as an offset transformation during IK. That is, after the static trial, the initial orientations in the IK analysis module are set to the transformation $R_{heading}{}^{G_{o}}R^{G_{IMU}}{}^{G_{IMU}}R^{S_{i}}\left(t_{init}\right)$, while during the dynamic trial, the transformations passed to the IK module are $R_{heading}{}^{G_{o}}R^{G_{IMU}}{}^{G_{IMU}}R^{S_{i}}\left(t\right)$.

In our experiments, we made use of a custom-build IMU MOCAP system (based on NGIMU from x-io Technologies Limited) to record the upper- or lower-limb movement and test the real-time system. Unfortunately, this introduced many technical challenges related to obtaining accurate orientation from multiple sensors that are out of the current study’s scope. We experienced problems with the accumulation of errors due to bias and drifting in the orientations, which are sensitive to magnetic interference. Synchronization of sensor data (e.g., IMU, pressure insoles) was accommodated through a data structure that stores the arriving frames, which are then resampled (constant sampling frequency) using each sensor’s synchronized timestamp. Timestamp synchronization was performed at the beginning of each session. Despite the technical challenges, we were able to test whether one could successfully use IMU information as an input to our system and verify the Inverse Kinematics (IK) method.

### 2.2. Real-Time Filtering

Real-time filtering is a critically important step in our pipeline, influencing the quality of the obtained results from dynamic analysis. Indeed, since we are interested in performing ID analysis, one has to determine the first and second derivatives of the kinematic data (IK). Calculating derivatives is a challenging task in the presence of noise. If the noise is permitted in the ID calculations, we will obtain unrealistic estimates of the joint moments because the acceleration term is multiplied with the inertia mass matrix. To cope with this issue, we usually apply a low pass filter (e.g., zero lag fourth-order Butterworth filter [23]) and then use splines (e.g., generalized cross-validation splines [27]) to approximate the derivatives. To achieve good results, we typically record the full trial and then perform the filtering offline, considering past and future (non-causal filter) samples to calculate the current value. However, in a real-time setting, one receives the data frame sequentially. If we want to consider future samples, then we are technically introducing artificial delay (lag). We will show a trade-off between accuracy and computational latency and optimally determine the filter parameters as per task.

The proposed filter (Figure 2) assumes that the data frame arrives one by one at a constant sampling frequency ($f_{s}$) and is placed into a multi-dimensional circular buffer. Each time a new frame arrives, it is appended at the end of the circular buffer, discarding the oldest frame. In our implementation, the size of the circular buffer *M* is termed as the memory of the filter. A low-pass ($f_{c}$ cutoff frequency) Finite Impulse Response (FIR) filter of order *M* (same as memory size) is applied to reject the high-frequency noise. The filter’s kernel and window functions are sinc and Hamming, respectively. The filtered signal is used to construct a generalized cross-validation spline of order $N_{s}$. Consequently, they are used to calculate the first and second derivatives of the signal. We evaluate the spines at $t_{d}=t-D/f_{s}$ (*D* is the lag of the real-time filter) instead of *t* to improve the filtering result. Therefore, the proposed filter is characterized by four hyper-parameters (*M*, $f_{c}$, $N_{s}$, and *D*). Depending on the application and type of task, one can fine-tune these parameters to achieve the right trade-off between delay and accuracy.

### 2.3. Estimation of Ground Reaction Wrenches

To perform dynamic analysis, one must estimate the external GRF&M and include them in the calculations. These are typically measured using dedicated equipment (e.g., force plate). However, this can significantly limit the scope and extent of any dynamic analysis only when such measurements are available. There have been numerous studies [18,19,20,21,22,23,24] that try to determine the GRF&M using only kinematic information. In the context of real-time applications, a compromise must be made between complexity and accuracy.

In this work, we implement a GRF&M prediction method, where the total forces and moments are computed using the Newton–Euler equations of motion:(2)$$\begin{array}{cc} f_{total} & =\sum_{i}^{N}m_{i}\left(a_{i}-g\right)\\ \tau_{total} & =\sum_{i}^{N}\left[I_{i}{\dot{\omega }}_{i}+\omega_{i}\times \left(I_{i}\omega_{i}\right)\right]-\sum_{i}^{N}\sum_{j}^{K_{i}}\left(r_{ij}\times f_{ij}\right) \end{array}$$
where $f_{total}$ denotes the total external force, *N* the total number of body segments, $m_{i}$ the mass of each segment, $a_{i}$ the linear acceleration of the center of mass, and g the gravitational force. In the second equation, $\tau_{total}$ is the total external moment, $I_{i}$ the inertia tensor with respect to the segment’s center of mass, and $\omega_{i}$, ${\dot{\omega }}_{i}$ the angular velocities and angular accelerations, respectively. $f_{ij}$ describes the forces applied to the $K_{i}$ end-points of each body segment with position vectors between the center of mass and the end-points denoted by $r_{ij}$.

Assuming the GRF&M are the only external forces and moments are applied to the subject during gait, the $f_{total}$ and $\tau_{total}$ are equal to the sum of the forces and moments applied to each foot:(3)$$\begin{array}{cc} f_{r}+f_{l} & =f_{total}\\ \tau_{r}+\tau_{l} & =\tau_{total}. \end{array}$$

During the single support phase, the GRF&M applied to the limb in contact with the ground is equal to the total GRF&M computed. Whereas, during the double stance phase, the indeterminacy problem is resolved following the smooth transition assumption [18]. In this assumption, the transition functions $f_{t}$ are expressed in a walking direction coordinate system. The estimated forces and moments, which are expressed in the global reference frame, are transformed after computing the walking direction from the average orientation of the pelvis segment’s anterior axis. This results in two new vectors $f_{total}^{w}$ and $\tau_{total}^{w}$.

An import prerequisite for using the transition functions is determining time events during the gait cycle. In particular, the heel-strike $t_{hs}$ and toe-off events $t_{to}$ as well as the time periods $T_{ds}$ and $T_{ss}$ of the double and single support phase, respectively. In this work, we implemented a family of methods that share a common architecture presented in the state diagram (Figure 3). State changes occur when selected values such as the magnitude of generated contact forces in proximity with virtual ground surfaces, estimated acceleration values of lower limb body segments based on kinematics, or equivalent measurements from external sensors (pressure insoles or accelerometers). This information is used to determine the gait state (*stance* or *swing*) based on a predefined threshold value $v_{th}$. The gait events and periods are determined online by monitoring changes in the last *k* consecutive states. The *leading leg* is assigned to the lower limb corresponding to the last heel-strike event until the next heel-strike event of the contralateral leg. After computing the gait-event-related parameters, the GRF&M applied on each lower limb are calculated following the rules defined in Table 1.

The CoP is approximated as described in [28]. We scale the distance d, from the heel to the metatarsophalangeal joint during the single stance phase with the stance frequency $\omega =2\pi /T_{ss}$ using the following scaling factor:(4)$$\sigma =-\frac{2}{3\pi }\left(\sin\omega t-\frac{\sin2\omega t}{8}-\frac{3}{4}\omega t\right)\in \left[0,1\right].$$

The computed vector σ·d is projected on the virtual plane representing the ground. After determining the external wrenches, we perform an ID calculations using the recursive Newton–Euler method, which is very efficient.

### 2.4. Real-Time Estimation of Muscle Forces

Estimation of muscle forces imposes several challenges because there is not a unique solution due to the muscle redundancy problem [29] (more unknowns than known equations). Unfortunately, we can neither measure muscle forces nor measure the activity of all muscles (e.g., deep muscles). Therefore, optimization is one of the few viable methods that one could use to determine these unknowns, which are essential for consequent calculations (e.g., joint loads) and different applications. It has been shown that the central nervous system is controlling the muscles in a manner that minimizes the effort during cyclic movements in healthy subjects [30]. Objective criteria such as muscle stress squared or activation squared seem to explain well experimental recording of muscle activity [31].

In our implementation, we formulate the optimization problem as follows:(5)$$\begin{array}{cc} \underset{f_{m}}{\mathrm{minimize}}\; & \frac{1}{p}\sum_{i}^{M}{\left(\frac{f_{m,i}}{f_{m,i}^{max}}\right)}^{p}\\ \mathrm{subject}\;\mathrm{to}\; & \left[\begin{array}{c} 0_{6\times 1}\\ \tau_{N-6\times 1} \end{array}\right]=\left[\begin{array}{c} 0_{6\times M}\\ R{\left(q\right)}_{N-6\times M} \end{array}\right]f_{m}\\ \; & f_{m}⪰0 \end{array}$$
where $f_{m}\in R^{M}$ denotes the muscle forces, $f_{m}^{max}\in R^{M}$ the muscle maximum isometric force, *p* is the power exponent (set to 2 or 3), $R\in R^{(N-6)\times M}$ the moment arm matrix that depends on the generalized coordinates ($q\in R^{N}$), and $\tau \in R^{N-6}$ the generalized forces. The residual forces of the pelvis are not included in the above optimization, because they are a byproduct of modeling inconsistencies [9]. The above formalization requires that we have already solved the IK and ID problems to compute q and τ, respectively. Please note that the inequality constraint $f_{m}⪰0$ ensures that the muscles can only pull, but it does not restrict the upper bound.

Solving Equation (Equation 5) in real-time is challenging because it takes time to evaluate the moment arm matrix and solve the nonlinear optimization in fewer steps. To this end, the moment arm matrix is approximated symbolically. To derive a symbolic representation, multivariate polynomial fitting [7,32] was performed on samples of the muscle moment arm at different configurations. To reduce the complexity and improve the fit’s robustness, we determined the coordinates affecting each element in the moment arm matrix by identifying the DoF spanned by each muscle. To speed up the optimization, we can control the convergence tolerance and “warm-start” from the previously obtained solution. We used the interior point algorithm [33] to solve the nonlinear, constrained optimization problem.

### 2.5. Developing a Gait Retraining System

In the previous subsections, we presented a pipeline for estimating the kinematics, dynamics, and internal forces (muscle and joint) of the musculoskeletal system in real-time. The pipeline serves as the backbone (back-end) for implementing rehabilitation solutions that could provide real-time feedback to the user and clinicians (instructor). In this subsection, we present a gait retraining system (front-end) for reducing the knee joint’s loading and reduce the pain. The gait retraining system aims to engage the users in a series of exercises that can teach them how to adapt their gait optimally.

The gait retraining system was implemented using the Unity3D game engine. Unity3D was selected because it can permit the implementation of solutions that can target typical projection screens and utilize augmented or virtual reality output devices. Our system could support interactive scenarios that could engage the user and make the training process more fun and effective. Exchange of data with the simulation back-end was established through shared memory to improve efficiency, reduce latency, and decouple implementation specifics (e.g., simulation back-end written in C++ and front-end in C#).

Figure 4 presents the front-end main window. It contains the musculoskeletal visualizer, real-time plotting, and footprint visualization. Within the application, the operator can monitor the logged users’ progress, replay sessions, analyze events, and create new objectives and scenarios. The expert can define a session’s objectives based on values obtained from the back-end, namely foot progression angle, vertical knee joint reaction forces, knee adduction moment, and trunk angle. Once the range and desired values are defined, we can use intuitive visualization primitives such as bar, gauge, or text indicators to provide real-time feedback.

In Figure 5, we present the gamification approach used to engage the user during a train session. The users gather points according to their performance, following the clinician’s objectives during a session. A combo mechanism is used to motivate and encourage users to follow the instructions and maintain the changes throughout the session’s whole duration. Each session, the user is awarded the gathered and golden points calculated based on how well the patient followed the suggestions. Users can monitor their performance on the dashboard, examine key statistics and significant milestones that reflect their progress.

The extension of this system is not only limited to the particular case presented here. One can imagine that similar rehabilitation scenarios could be implemented in line with the applications’ needs. The significance of the back-end is that it permits calculating any kinematic and dynamic quantity (joint angles, moments, muscle forces, reactions) from the musculoskeletal model in real-time. Depending on the application, different combinations of these quantities will be needed. The front-end can then transform the raw information and visualize it appropriately. The user’s interaction may not be only limited to primitive projection screens but can also utilize haptic, augmented, and virtual reality devices. Therefore, rehabilitation scenarios could be designed so that they are more engaging and effective. All this would not be possible without an accurate and reactive back-end.

## 3. Results

In this section, we present the comparison of the real-time filtering method’s performance and justify how to determine the different hyper-parameters. Next, the comparison of the offline and real-time generalized forces computed through ID is presented by highlighting the importance of determining the kinematics’ first and second derivatives. We will further present the estimated GRF&M and quantify their impact on the ID results. A comparison between the offline and real-time calculations of muscle forces is examined. Computation delays for the various modules are presented in the text. Finally, we outline a use case scenario of the proposed system in the context of rehabilitation and gait retraining to reduce the knee’s joint reaction loads.

Our framework utilizes the OpenSim’s API [9,10,34] and all comparisons are made against its offline methods for kinematic and dynamic calculations. The musculoskeletal model used in this study is based on gait2392 [8] with some minor modifications. The DoF have been reduced to 19 and the model is actuated by 92 Hill-type muscles [35]. The generic model is scaled using a static trial to account for the subject’s anthropometry. Experiments were carried out on Intel(R) Core(TM) i5-3320M CPU @ 2.60GHz. Better CPU specs can further reduce the computational delays.

### 3.1. Performance of Real-Time Filtering and
Differentiation

In this subsection, we demonstrate the real-time filter’s performance and outline the procedure for determining the optimal values of its parameters. The filter operates on the generalized coordinates obtained from IK since the optimization may not result in a smooth solution. The mean computational delay of the IK module is less than 0.1 ms. Four parameters, namely the cutoff frequency $f_{c}$, spline order $N_{s}$, memory size *M*, and delay *D*, characterize the filter. The type of movement determines the cutoff frequency. For slow movements like gait, we chose $f_{c}=6$ Hz as is the case in most studies [14,36]. We also fixed the spline order to $N_{s}=3$, a reasonable value to avoid over-fitting. Determining parameters *M* and *D* is discussed below.

The purpose of the filter is to reject the noise and calculate the first and second derivatives of the kinematics necessary for consecutive dynamic analysis. To determine the optimal values of the two hyper-parameters (*M* and *D*), we examined the total Root Mean Squared Error (RMSE) (Figure 6) of the filtered kinematics comparing the real-time filter and OpenSim’s offline kinematics analysis. The OpenSim’s kinematics analysis utilizes the same low-pass filter and spline method, having at its disposal the whole duration of the movement. We observed that for each type of error (coordinates, speeds, accelerations), there exists an optimal value for *M*. We determined that M=35 is the optimal choice for this movement because acceleration errors can significantly impact the consequent analysis. Besides, a small value of *M* reduces the computational latency of the filter. Similarly, D=14 seems to be the optimal solution, which can lead to an artificial lag of 0.14 s for a typical marker sampling frequency of 100 Hz. Notably, one can follow this procedure to determine the optimal parameters for any movement.

To highlight the influence of filtering on the joint angles and derivatives, we compared (Figure 7) OpenSim’s offline kinematics analysis (ground truth) against the proposed filter and the “spatial” filter as implemented in rtosim [14]. The only parameter of the spatial filter is the cutoff frequency, which we set to 6 Hz. We can observe that the proposed filter has a lower RMSE and closely resembles OpenSim’s offline time-series. While the spatial filter can approximate the original signal well, it underestimates its first and second derivatives for fast-moving coordinates (comparison of all coordinates presented in the Appendix A). The spatial filter has low computational latency (mean latency less 0.1 ms) than the proposed filter (mean latency of 3 ms) and is very easy to implement. However, acceleration errors can lead to significant differences during dynamic analysis (presented in the next subsection).

### 3.2. Influence of Online Filtering on Inverse Dynamics
Calculations

We will briefly discuss the performance of the real-time ID module with respect to OpenSim’s offline counterpart. The ID module implements a recursive Newton–Euler formulation, which is very efficient, achieving a mean computational latency of less than 0.1 ms. Please note that the OpenSim’s ID offline analysis has at its disposal the full trial in advance and thus can apply non-casual operations to reject the noise and calculate derivatives. In contrast, the real-time system operates on the arriving frames and must process a limited number of instances (filter’s memory buffer) to reduce latency while maintaining accuracy.

Moreover, we present the effect of online filtering on the calculated generalized forces from ID and why proper filtering is important. In Figure 8 the comparison of the offline ID from OpenSim and real-time ID utilizing the two real-time filters presented previously (proposed and spatial) is illustrated. A comparison of all coordinates is also presented in Appendix A. The time lag introduced by the filters was compensated (shifted) in these plots for comparison reasons. We observed that the real-time results closely match the offline ones. Residual vertical forces at the pelvis are low and within the accepted norm [37]. The initial part of the proposed filter’s curve is missing because we need at least *M* samples to begin the processing. The real-time curves are smoother and present fewer oscillations due to the non-causal implementation related to the *D* parameter. We also observed that the proposed filter outperforms the spatial filter in terms of RMSE. The spatial filter can underestimate the moments due to the kinematic mismatch at the coordinate velocity and acceleration, as presented in the previous subsection. Therefore, this can have a significant impact on the joint moments.

### 3.3. Prediction of Ground Reaction Wrenches and Influence on Joint
Moments

Predicting ground reaction wrenches in real-time can enable applications that calculate dynamic quantities (e.g., moments and muscle forces) during movement. In Figure 9, we present the comparison of the estimated and measured GRF&M during walking. The forces in the forward (*x*) and vertical (*y*) directions closely resemble the measured ones. Lateral (*z*) forces follow the overall shape. The *x* component of the CoP was accurately predicted, but the lateral *z* direction is hard to determine. Less confluence is observed in the moments. The mean computation delay of this module is 2 ms. The double support phase during walking leads to infinite possible solutions (close kinematic chain). Modeling assumptions, such as the distribution of mass and joint parameterization, can affect results due to the method’s nature.

Next, we quantified the influence of the predicted GRF&M on joint moments calculated from ID. In Figure 10, the comparison of the moments of the hip (flexion, rotation), knee, and ankle joints are depicted. Even though the predicted GRF&M closely resemble the measured ones, small discrepancies can significantly affect these moments. This becomes evident as we move from distal to proximal segments (e.g., errors at the ankle are smaller than at the hip). In the same figure, we also demonstrate which component of the estimated GRF&M influences the particular joint moments. In this context, we assumed that we perfectly know only either the CoP, CoP and moments, or CoP and forces. Notably, if we can correctly determine the CoP, then the joint moments’ mismatch at the ankle and knee is smaller. To improve the estimation of hip flexion moment, one must accurately determine both the CoP and forces. The ground reaction moments influence the hip internal rotation. These findings suggest that joint moments are sensitive to estimates of the GRF&M, but some joint moments can be determined accurately if one or two of the unknown variables are measured.

### 3.4. Comparison of Muscle Forces Determined in
Real-Time

In this section, the performance of the real-time muscle optimization module is presented. The determination of muscle forces can find many applications in sports biomechanics, ergonomics, and rehabilitation. For example, estimation of muscle forces is necessary to determine the joint reaction loads affected by muscles spanning the joints. Resolving the muscle redundancy is the bottleneck of the whole pipeline requiring an average of 10 ms to execute with a musculoskeletal model with 92 muscles. Things that have improved the performance of the muscle optimization were (i) the formulation of the problem to reduce the constraints and decision variables, (ii) the analytical representation of the moment arm matrix (precalculated), (iii) optimization initialized using the previous solution, and (iv) the efficiency and parallelism of the interior point optimization method.

Figure 11 presents the comparison between OpenSim’s Static Optimization (SO) and proposed real-time muscle optimization method. Results show forces for major muscles (gluteus maximus, semimembranosus, psoas major, rectus femoris, vastus medialis, biceps femoris short head, medial gastrocnemius, soleus, and tibialis anterior) of the lower-limb. A comparison of all 92 muscles is presented in the Appendix A. Good agreement between timing and shape matching is observed even though the two methods use different formalization and implementation specifics.

### 3.5. A Gait Retraining System for Reducing Knee Loads

In this subsection, we would like to highlight the usefulness of the proposed system in the context of gait retraining and rehabilitation. We will outline a possible use case scenario that might appeal to researchers and clinicians in the field of rehabilitation.
Dr. Good has experience with the gait retraining system, and their clinic just received a patient that has been diagnosed with mild knee osteoarthritis in her right leg. He already knows that the patient needs to adopt a gait pattern that alleviates the knee joint’s medial loading to reduce the pain. The primary indicator to assess the loading of the joint is the medial reaction force at the knee. One strategy is to adjust the foot progression angle, defined as the angle between the line of walking progression and the foot’s longitudinal axis. Therefore, he defines the vertical reaction force as an objective to be minimized. The decision variable is the foot progression angle. However, he also decides to experiment with the step length and step width, providing simple and intuitive visual feedback to the patient. The system can adjust the decision variables’ target value (Figure 12) using online gradient descent based on the joint reaction forces’ estimates.
First, he tried to familiarize the patient with the motion analysis lab by explaining the different visualization elements and by presenting the current session’s goal. Since this is her first session, the patient is ordered to walk normally to gather information on her walking habits and calibrate the model. The clinician checks if the system works properly by comparing the reaction loads’ real-time and offline calculations at the knee (Figure 13). Based on the calibration trial, he then sets the initial target values for the decision variables. The patient is instructed to walk by alternating her foot progression angle, step length, and step width based on the system’s indication. After exploring different gait modifications using real-time feedback, an optimal walking strategy is reached, resulting in reduced reaction loads on the knee’s medial compartment and is also comfortable for the patient. Both Dr. Good and the patient feel satisfied by the experience of using the gait retraining system, and the next appointment is scheduled.

## 4. Discussion

In this work, we presented the implementation and validation of a system capable of calculating various kinematic and dynamic quantities from a musculoskeletal model in real-time. We rigorously proved that the real-time calculations closely resemble their offline counterpart. This permitted us to examine the use of such a system for gait retraining to provide real-time biofeedback to the user. The added value of a gait retraining system needs to be further examined in the long-run, especially for reducing joint reaction loads in patients with osteoarthritis [25,38]. Nevertheless, we are one step closer to utilizing new biofeedback types such as predicted muscle forces or joint reaction loads in real-time and design clinically relevant scenarios through this work.

Being able to accurately determine the kinematics using either a marker- or IMU-based analysis systems is an important step that, if not done correctly, can affect calculations at the next stages. On the one hand, marker-based solutions are well-established and can accurately determine the movement of the reflective markers. However, in real-time scenarios, there might be instances of missing markers due to occlusion [26], which must be handled before processed by IK. On the other hand, IMU-based solutions can be utilized in out-of-the-lab settings permitting a new dimension of movement analysis and possible applications. Nevertheless, the limitation of IMU technologies lies in their accuracy, aggregation of errors (bias or drift), magnetic interference, and difficulties in calibrating their placement on the body. The synchronization of multi-unit sensors that communicate through the network is fundamental for the proper function of the system [39]. We addressed this by synchronizing the timestamp of each device at the beginning of the acquisition session. Arriving data frames were placed into a data structure that efficiently resamples the heterogeneous entries based on the synchronized timestamp information. In this manner, we could extend and support a multitude of input devices (e.g., pressure insoles or surface EMG) within the proposed framework. We were able to test whether one could successfully use IMU information as an input to our system and verify the pipeline. However, we could have benefited from utilizing a well-established IMU MOCAP suit such as XSENS instead of our custom-build solution and compare the performance of the system against marker-based solutions.

In order to perform dynamics calculations, we must determine the first and second derivatives of the kinematics. This proved to be a challenging task for real-time applications because of the inherent noise in the measurements that can easily pollute the meaningful signal. This is not an issue in an offline setting because we can calculate the value of a signal at *t* using prior or subsequent information. Using subsequent information dramatically improves the filtering process without distorting the original signal. This is essentially what our proposed filter does. It introduces a small lag (e.g., 14 samples) in the data to predict the signal more accurately. Moreover, we use splines instead of finite differences to calculate the derivatives. If the motion capture operates at higher frequencies (e.g., 100 Hz), then a small delay will be negligible to the user. The drawback is that this filter can slightly increase the computation time and thus reduce the pipeline’s throughput. However, the benefits are that the kinematics are accurately determined, and therefore real-time filtering does not distort calculations at the following stages. It is also worth noting that one could determine the optimal filter parameters for a given type of task, such that they minimize the inconsistencies between the offline and online analysis, as shown in this study.

Estimating the GRF&M in real-time under any movement context is an essential step towards moving in out-of-the-lab settings. This holds for any interaction of the user with the environment. In this study, we used a model-based approach to estimate the GRF&M based on first principles. While these wrenches approximate the measurements well, we observed that even a small mismatch could easily manifest in the calculated joint moments. Interestingly, some components of the ground reaction wrench influence these moments differently. A model-based approach might be challenging to implement and probably hard to generalize to different types of movements. Furthermore, more accurate models might result in further computation latencies. Given that rich data sets of movements can be recorded, we think that a machine learning method might appeal better in the context of real-time applications.

Predicting muscle forces are an essential aspect of utilizing musculoskeletal models because direct measurements are challenging and sometimes impossible. Invasive techniques raise many issues, whereas the mapping of muscle activity to muscle force is not trivial. Non-invasive assessment of muscle-tendon loads through vibration tracking [40] is a promising technology currently at a development stage. Well-validated and calibrated musculoskeletal models can provide us with reasonable estimates of the internal forces for cyclic movements (e.g., walking and running) even though one needs to resolve the muscle redundancy problem [31]. Determining the muscle forces in real-time is the most computationally expensive operation presented in this pipeline. To alleviate this, we sought to formulate the optimization in a manner that is easy to solve while also precomputing specific computationally expensive quantities (analytical representation of the moment arm matrix). Formulating the optimization in terms of muscle forces allowed us to remove the upper-bound on the decision variables and avoid including reserve actuators (additional unknowns) to compensate for the model’s inconsistencies. The latter is the typical case when one optimizes for muscle activations ($0⪯a_{m}⪯1$) because the muscles may not be able to produce the right amount of force to satisfy the movement. The number of constraints (six equality constraints) was further reduced by avoiding optimizing for non-physiological forces applied on the pelvis. Importantly, we showed that the predicted muscle forces closely resemble the ones obtained from a typical SO formulation in OpenSim. The limitation of the current implementation lies in the optimization routine’s ability to solve the problem efficiently. Nevertheless, we are exploring how to improve on this and possibly include additional constraints that could originate from direct measurements such as surface EMG to personalize the obtained solutions optionally.

Calculation of muscle forces permits the estimation of joint reaction loads. The plethora of gait retraining systems rely on estimating the knee adduction moment using the ID method to meet applications’ real-time constraints [41,42,43]. The knee adduction moment is a measure which if reduced, can ease knee pain for subjects that are affected by osteoarthritis in the medial cartilage compartment [43]. However, this measure cannot predict the effect of muscle co-contraction and its role in stabilizing the joints and regulating their load [14]. Obtaining joint reaction loads are computationally expensive due to muscle optimization. However, it provides us with richer information that could help design new rehabilitation solutions.

## 5. Conclusions

In this work, we presented a framework for calculating a multitude of kinematic and dynamic quantities using musculoskeletal models in real-time. Such a system can find application in rehabilitation by providing biofeedback to the users during the training sessions. Notably, the system is designed to work with marker- and IMU-based MOCAP solutions. The second allows exploring scenarios where users’ habits can be monitored and analyzed outside a dedicated and well-equipped laboratory (outdoor activities). The out-of-the-lab setting is still an ambitious goal that is only possible for kinematic analysis. Performing dynamic analysis without additional sensing devices is challenging due to the complexity of modeling any user’s interactions with the environment (e.g., ground reactions, contacts with objects). Within this vision, we think that the future lies in this direction, where interactions can be accurately predicted, and kinematic and dynamic analysis can then be performed in real-time. This technological leap will unveil new research directions that could target not only applications related to healthcare and human well-being but also translate to other fields as well. We hope that the methods outlined here and the publicly available source code can help meet this grand challenge.

## Figures and Tables

**Figure 1 sensors-21-01804-f001:**
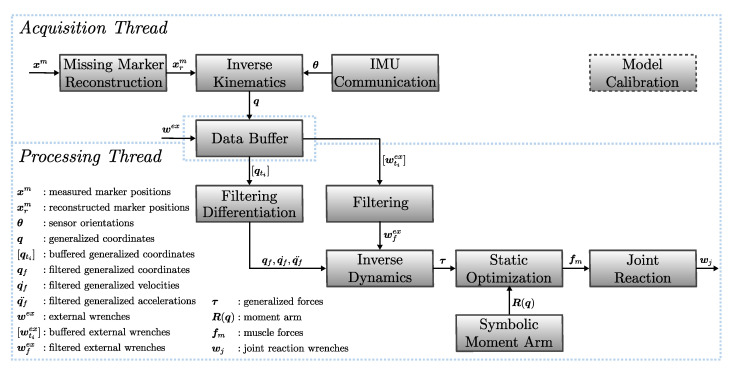
Overview of the real-time analysis pipeline. The acquisition thread collects raw data from sensor inputs and performs IK to obtain the kinematics from marker- and IMU-based sources. Data are shared using thread-safe data structures. Depending on the application, the processing thread can arrange the operations required to extract meaningful information from the musculoskeletal model in real-time. Model calibration refers to the process of adjusting the musculoskeletal model to account for the subject’s anthropometric parameters using static trials and functional tasks.

**Figure 2 sensors-21-01804-f002:**
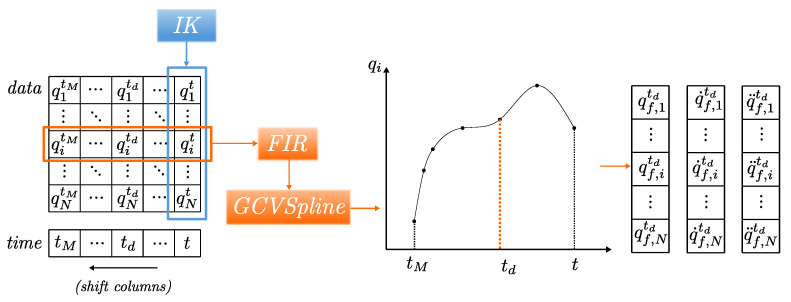
An overview of the filtering and differentiation module. The generalized coordinates from IK are placed into a circular buffer of memory *M*. The *M* samples of each signal are filtered using a low pass FIR filter. A generalized cross-validation spline of order $N_{s}$ is then constructed to evaluate the coordinate and derivatives’ value at time instance $t_{d}$.

**Figure 3 sensors-21-01804-f003:**
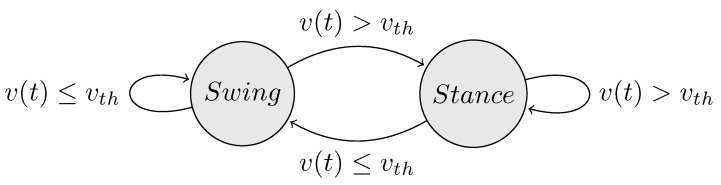
State diagram denoting the conditions required to determine the current gait state. Transitions between *swing* and *stance* phases occur by comparing the selected measure u(t) with a threshold value $u_{th}$.

**Figure 4 sensors-21-01804-f004:**
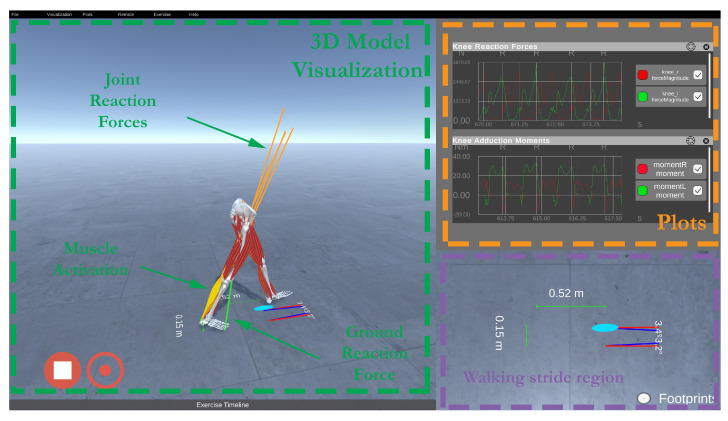
Visualization front-end composed of a musculoskeletal visualizer, real-time plotting, and footprint visualization.

**Figure 5 sensors-21-01804-f005:**
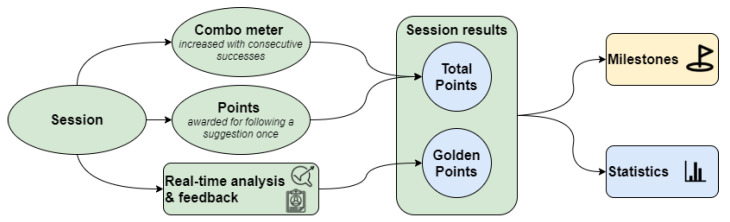
Gamification approach based on session awarded points. Users can monitor their performance on the dashboard, examine key statistics and major milestones that reflect their progress.

**Figure 6 sensors-21-01804-f006:**
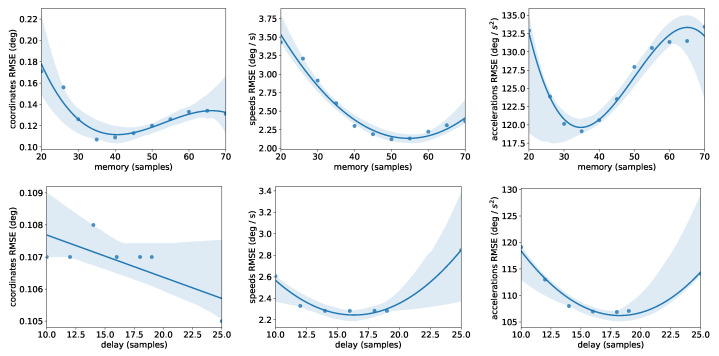
RMSE between OpenSim’s offline kinematics analysis and proposed filter as a function of the decision variables (*M* and *D*). We selected the smallest values that resulted in the lowest error in the acceleration.

**Figure 7 sensors-21-01804-f007:**
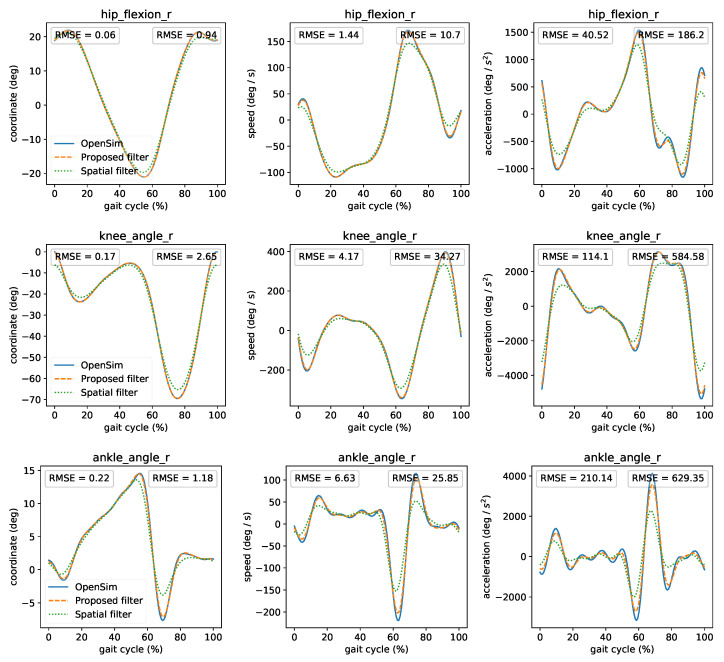
Comparison between OpenSim’s offline kinematics analysis, the proposed, and spatial filters. Left RMSE compares OpenSim’s and proposed filter’s kinematics. Right RMSE compares OpenSim’s and spatial filter’s kinematics. For comparison reasons, we shifted the curves to compensate for the time lag introduced by the filters.

**Figure 8 sensors-21-01804-f008:**
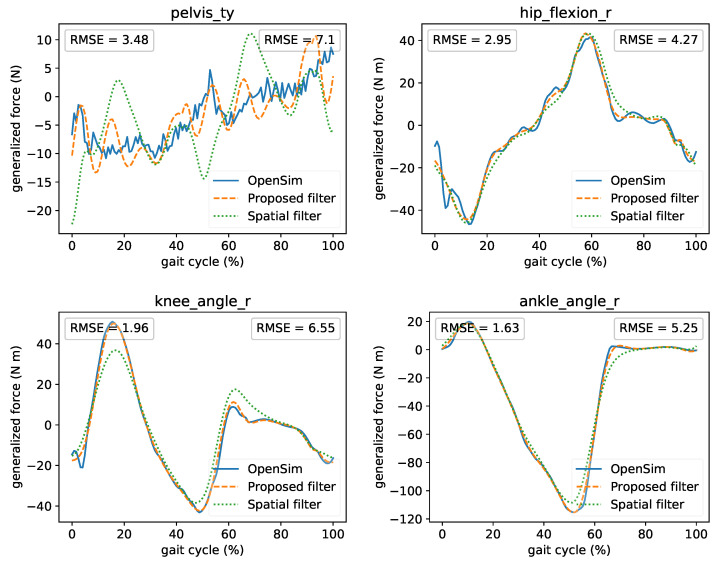
Comparison among offline ID from OpenSim and real-time ID using the two real-time filters. Left RMSE compares OpenSim’s ID and real-time’s ID that uses the proposed filter. Right RMSE compares OpenSim’s ID and real-time’s ID that uses the spatial filter. For comparison reasons, the curves were shifted to compensate for the time lag introduced by the filters.

**Figure 9 sensors-21-01804-f009:**
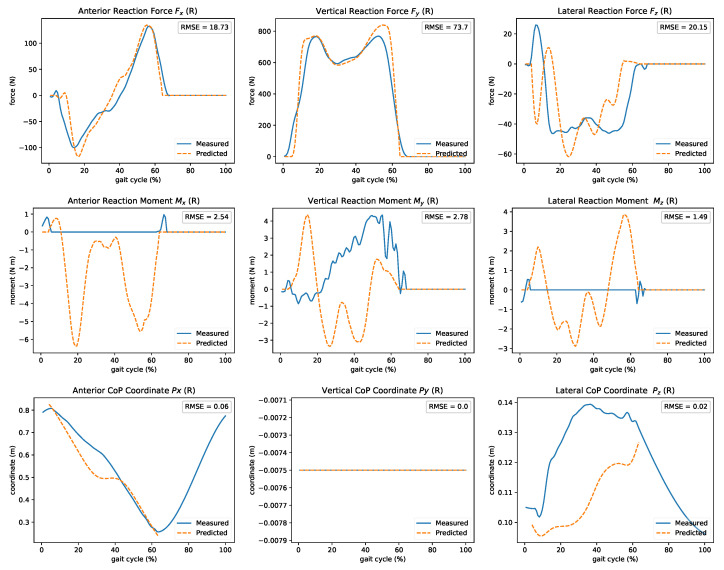
Real-time estimated ground reaction forces, moments, and CoP during gait. Good agreement of the predicted forces, especially on the forward (*x*) and vertical (*y*) directions. In contrast, only the *x* component of the CoP was accurately predicted. Less confluence is observed in the moments.

**Figure 10 sensors-21-01804-f010:**
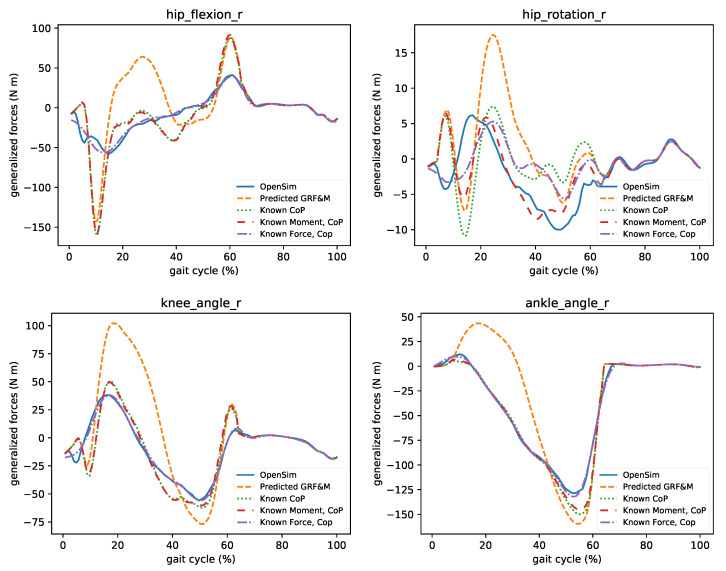
The influence of estimated GRF&M on the generalized forces computed through ID. Predicted GRF&M greatly affect the calculated joint moments. A good estimate of CoP improves the predictions of ankle and knee moments. Hip flexion requires accurate estimation of both CoP and forces, while hip rotation CoP and moments.

**Figure 11 sensors-21-01804-f011:**
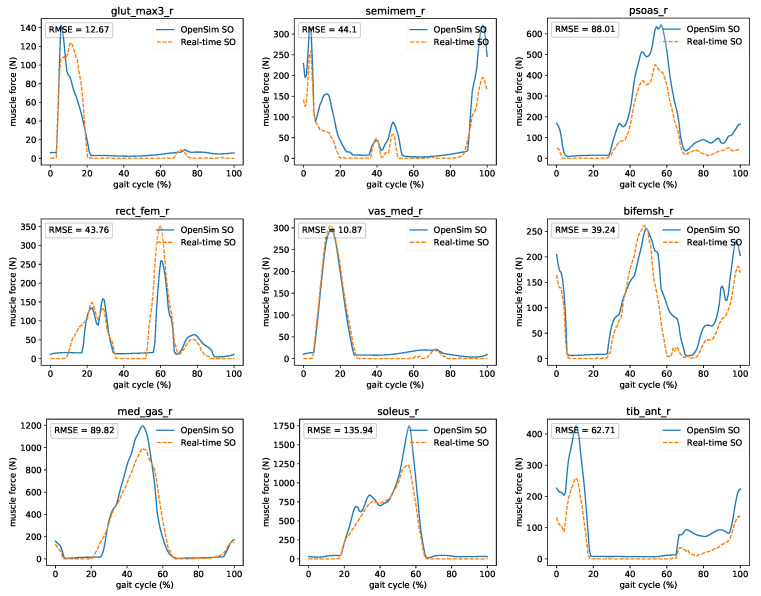
Comparison between OpenSim’s SO and proposed real-time muscle optimization method. The force of major right leg muscles: gluteus maximus, semimembranosus, psoas major, rectus femoris, vastus medialis, biceps femoris short head, medial gastrocnemius, soleus, and tibialis anterior was presented.

**Figure 12 sensors-21-01804-f012:**
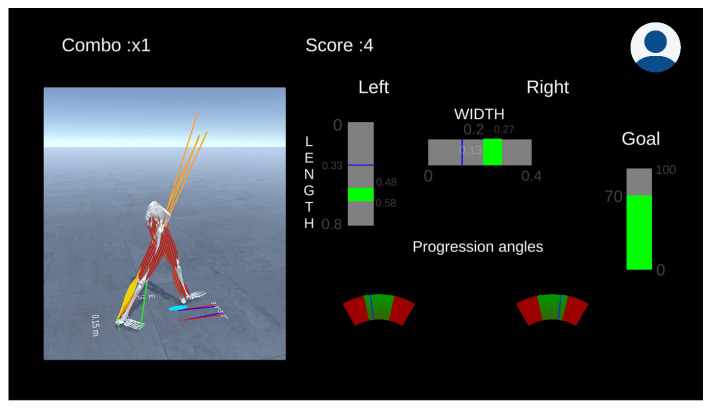
The gait retraining session aims to reduce knee reaction loads by adjusting the foot progression angle, step length, and step width (decision variables). The green regions of the decision variables are adjusted online according to the reaction forces’ estimation on the diseased knee. The user is awarded points if the instructions are followed.

**Figure 13 sensors-21-01804-f013:**
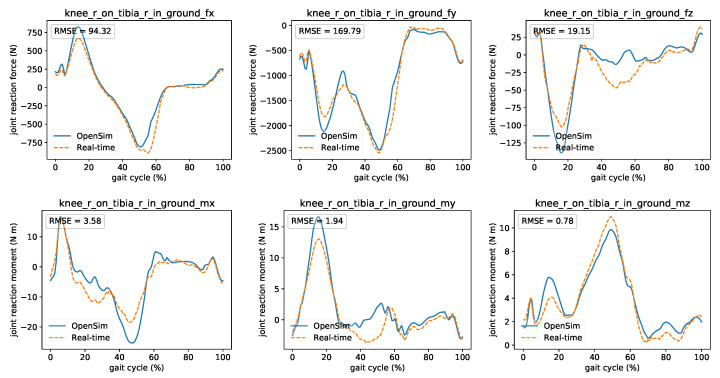
Comparison between OpenSim’s (offline) and real-time’s joint reaction loads (forces and moments) on the knee using the muscle forces obtained by the proposed optimization. For comparison reasons, the curves were shifted to compensate for the time lag introduced by the filter.

**Table 1 sensors-21-01804-t001:** According to the gait phase, classification of calculations for determining the forces and moments applied on the leading and trailing legs. During the single support phase, the GRF&M of the limb in contact with the ground are equal to the total external force $f_{total}^{w}$ and moment $\tau_{total}^{w}$. Whereas, during the double support phase, the results are distributed between the two legs using the transition functions $f_{t}\left(t\right)$ described by [18]. The ⊙ operator is used to denote element-wise multiplication.

Reaction Component	Double Support	Single Support
$f_{trailing}^{w}\left(t\right)$	$f_{total}^{w}\left(t_{hs}^{leading}\right)⊙f_{t}\left(t\right)$	0
$m_{trailing}^{w}\left(t\right)$	$\tau_{total}^{w}\left(t_{hs}^{leading}\right)⊙f_{t}\left(t\right)$	0
$f_{leading}^{w}\left(t\right)$	$f_{total}^{w}\left(t\right)-f_{trailing}^{w}\left(t\right)$	$f_{total}^{w}\left(t\right)$
$m_{leading}^{w}\left(t\right)$	$\tau_{total}^{w}\left(t\right)-m_{trailing}^{w}\left(t\right)$	$\tau_{total}^{w}\left(t\right)$

## Data Availability

The data supporting the reported results can be found on https://github.com/mitkof6/OpenSimRT along with the code that generates the figures.

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
