# Peer review of "Real-Time Musculoskeletal Kinematics and Dynamics Analysis Using Marker- and IMU-Based Solutions in Rehabilitation"

_sensors, 2021, doi:10.3390/s21051804_

Round 1

Reviewer 1 Report

Authors presented an open-source framework for carrying real-time calculations of various musculoskeletal kinematic and dynamic quantities using marker-based and IMU-based system.

Marker-based solutions are well established and can accurately determine the movement of reflective markers, whereas occlusion can be problematic in real-time scenarios due to missing makers. From this point of view, the proposed system can provide real-time analysis results rather than offline analysis, so it can be used in various application fields that can provide real-time feedback to users during training sessions.

The authors presented the problem clearly and systematically demonstrated the proposed method through experiments. However, the following should be modified for the author's understanding.

  1. In general, a low-order IIR filter is used as a real-time low pass filtering to obtain a minimum time delay. However, for low frequencies such as human motion, it is an appropriate approach to use a high-order FIR filter as suggested by the authors.
    As specified in Figure 6 and the main text (line 306), the buffer size (M) is 35, and if the current input value is put into the buffer after the operation is completed, the order of M and the FIR filter will be the same. Therefore, the order of the FIR filter of line 156 is M, not M/2, so it needs to be confirmed.
  1. The time delay of the N-order FIR filter is (N-1)/2, so it is (35-1)/2, which is 14 samples. Therefore, it is necessary to rewrite the sentence of line 308.
  2. You specified that the time lag for the marker frequency of 100hz is 0.14 ms, but applying the result of comment 2, it becomes 14 samples/100hz = 140 ms or 0.14 sec. Therefore, it is necessary to modify this part.

Author Response

Dear reviewer,

Thank you very much for reviewing our work.

Comment 1

Thank you for pointing this mistake. Indeed we set the order to be the same as the size of the memory buffer.

[lines 163-165] ... A low-pass ( fc cutoff frequency) Finite Impulse Response (FIR) filter of order M (same as memory size) is applied to reject the high-frequency noise. ...

Comment 2

Thank you for pointing out this formula. For M = 35, D = (35 - 1) / 2 = 17. However, from the experiments (figure 6, bottom), we see that we do not have a substantial reduction in the error compared to D = 14. Therefore, for the following experiments, we choose the smallest value because it gives us smaller latency.

Comment 3

Thank you for spotting the mistake in the units.

[lines 316-317] ... Similarly, D = 14 seems to be the optimal solution, which can lead to an artificial lag of 0.14 s for a typical marker sampling frequency of 100 Hz. ...

Reviewer 2 Report

Dear Authors,

The article is very well structured and provides appropriate introductory information. The presented methodology is scientifically sound and the conclusions are supported by the presented experimental results.

For this reason, it is my opinion that the manuscript is accepted for publication.

Author Response

Dear reviewer,

Thank you very much for reviewing our work.

Reviewer 3 Report

The work presents real-time assessment of kinematics and dynamics during movements using computational musculoskeletal models. This is based on a previous published work presenting a system capable of assessing human movement and muscle function in real-time.

The work is clearly presented, comprehensive and of high quality.

Specific comments:

Introduction:

The introduction presents adequately the problem, but the aim can be better presented. Concretely there are 2 sentences together that can be merged into one for clarification: Lines 26 to 30: “To this end, we aim to examine the possibility of estimating a multitude of

27 kinematic and dynamic quantities using computational musculoskeletal models in real-time for

28 both in- and out-of-the-lab conditions.

29 This study’s scope falls within the context of real-time assessment of kinematics and

30 dynamics during movements relying on computational musculoskeletal models.”

The rest of the paper is well described. 

Author Response

Dear reviewer,

Thank you very much for reviewing our work.

Thank you for your suggestion. Indeed the two sentences mean the same thing. We have corrected the second sentence (the first sentence of the second paragraph).

[lines 29-30] ... Different research groups have approached the problem of estimating the kinematics and dynamics of movement using musculoskeletal models. ...

Reviewer 4 Report

The paper contains a valuable contribution. The subject is within the scope of the journal and the objective of research is well stated.

In the opinion of this Reviewer the manuscript deserves to be published once the Author takes into account the raised issues.

Introduction / Literature review

  1. Even if the papers cited refer to EMG, in do the authors think that it would be more appropriate to talk about sEMG rather than EMG?
  2. For the sake of readability, at the end of Section 1 the authors should describe how the paper is structured.

Discussion

  1. I perfectly agree with your concerns about the synchronization and errors accumulation of the IMU, especially in an outdoor environment. The reference 39 is about a 6 DoF and not 9DoF that the authors need. Do the authors think that the proposed framework could use a multi-unit synchronized system for activity monitoring with IMU and sEMG capabilities (e.g., https://doi.org/10.3390/electronics9071118, https://doi.org/10.1109/ROBIO.2015.7418906, document that could be cited in the text)?

Minor

  1. Specify the symbolism used for the transposed vectors and matrix.
  2. The authors should check that the legends do not cover the graphs in the data.
  3. The authors should use N m (with the space) or N∙m instead of Nm.
  4. Mainly the English is good and there are only a few typos. However, the paper should be carefully rechecked.

Author Response

Dear reviewer,

Thank you very much for reviewing our work.

Comment 1

Thank you for raising this issue. We carefully examined the use of EMG in the text. When needed, we included surface or converted to muscle activity.

[lines 50-56] ... These methods take into account surface EMG measurements to calibrate the musculoskeletal model by adjusting muscle-related parameters while satisfying joint moments about one or more DoF. This approach's limitation is that surface EMG measurements can induce noise due to soft-tissue movement artifacts that can cause problems during calibration and use of the method. Besides, a superficial measurement is not an ideal representative of whole muscle function. ...

[lines 219-220] ... Unfortunately, we can neither measure muscle forces nor measure the activity of all muscles (e.g., deep muscles). ...

[lines 473-474] ... Invasive techniques raise many issues, whereas the mapping of muscle activity to muscle force is not trivial. ...

[lines 490-492] ...  Nevertheless, we are exploring how to improve on this and possibly include additional constraints that could originate from direct measurements such as surface EMG to personalize the obtained solutions optionally. ...

Comment 2

Thank you, we agree. The last paragraph (lines 75-91) of the introduction was restructured to include forward references of the sections.

Comment 3

Thank you for the interesting reading. Indeed, one of the major challenges that we faced was the synchronization of sensor inputs that could have different operation modes. In one of our demonstrations (demo video), we used IMUs to track the lower limb movement and pressure insoles to guide the ground reaction force prediction. We did not present this part because we wanted to carry out further experiments. The sensors' data frames were placed into a data structure, which is resampled efficiently based on each sensor's synchronized timestamp information. With this, we were able to fuse different sensors within the proposed framework. We did not experiment with sEMG measurements due to our initial design goal (use muscle optimization). Nevertheless, integration of additional input sources has been anticipated in our software design.

[lines 140-143] ...  Synchronization of sensor data (e.g., IMU, pressure insoles) was accommodated through a data structure that stores the arriving frames, which are then resampled (constant sampling frequency) using each sensor’s synchronized timestamp.  Timestamp synchronization was performed at the beginning of each session.  ...

[lines 437-446] ...  The synchronization of multi-unit sensors that communicate through the network is fundamental for the proper function of the system [39]. We addressed this by synchronizing the timestamp of each device at the beginning of the acquisition session. Arriving data frames were placed into a data structure that efficiently resamples the heterogeneous entries based on the synchronized timestamp information. In this manner, we could extend and support a multitude of input devices (e.g., pressure insoles or surface EMG) within the proposed framework. We were able to test whether one could successfully use IMU information as an input to our system and verify the pipeline. However, we could have benefited from utilizing a well-established IMU MOCAP suit such as XSENS instead of our custom-build solution and compare the performance of the system against marker-based solutions. ...

Comment 4

Thank you for the suggestion. We have used the T superscript notation for the transpose operation. It is explained after equation (1):

[lines 118-119] ... The superscript T over a vector or matrix denotes the transpose operation. ...

Comment 5

Thank you and sorry for this. We updated some of the figures manually so that the legends do not overlap the curves. The plots were generated automatically using scripts that compare the results for all model coordinates/muscles at different stages and not just those presented in the manuscript. It is not easy to ensure that legend or annotations do not overlap the curves. We also changed the transparency of the annotations (in all figures) to cope with this issue.

Comment 6

Thank you. We have updated the text and figures using N m (with space).

Comment 7

Thank you. The authors and colleagues carefully checked the text. We have highlighted the corrections in the revised manuscript.